# *Disrupted-in-schizophrenia-1* is required for normal pyramidal cell–interneuron communication and assembly dynamics in the prefrontal cortex

Jonas-Frederic Sauer*, Marlene Bartos*

Institute for Physiology I, Faculty of Medicine, University of Freiburg, Freiburg, Germany

**Abstract** We interrogated prefrontal circuit function in mice lacking *Disrupted-in-schizophrenia-1* (Disc1-mutant mice), a risk factor for psychiatric disorders. Single-unit recordings in awake mice revealed reduced average firing rates of fast-spiking interneurons (INTs), including optogenetically identified parvalbumin-positive cells, and a lower proportion of INTs phase-coupled to ongoing gamma oscillations. Moreover, we observed decreased spike transmission efficacy at local pyramidal cell (PYR)-INT connections in vivo, suggesting a reduced excitatory effect of local glutamatergic inputs as a potential mechanism of lower INT rates. On the network level, impaired INT function resulted in altered activation of PYR assemblies: While assembly activations defined as coactivations within 25 ms were observed equally often, the expression strength of individual assembly patterns was significantly higher in Disc1-mutant mice. Our data, thus, reveal a role of Disc1 in shaping the properties of prefrontal assembly patterns by setting INT responsiveness to glutamatergic drive.

*For correspondence:
jonas.sauer@physiologie.uni-freiburg.de (J-FS);
marlene.bartos@physiologie.uni-freiburg.de (MB)

Competing interest: The authors declare that no competing interests exist.

## Editor's evaluation

This article explores prefrontal cortex circuit function in mice lacking *disrupted-in-schizophrenia-1* (Disc1-mutant mice). a risk factor for psychiatric disorders. The data show specific impairment in the function of specific cortical interneuron populations, such as fast-spiking interneurons. Furthermore, it also showed a decreased spike transmission efficacy at local pyramidal cell–interneuron connections in vivo. The impaired interneuron function also resulted in altered activation of pyramidal cell assemblies.

## Introduction

Cognitive impairment is a major burden for individuals suffering from psychiatric disease. A series of investigations has put emphasis on the critical role of GABAergic interneurons (INTs) in the emergence of cognitive dysfunction in psychiatric illnesses (*Akbarian et al., 1995*; *Mirnics et al., 2000*; *Volk et al., 2000*; *Lewis et al., 2005*; *Uhlhaas and Singer, 2010*; *Lisman, 2012*; *Volk et al., 2016*; *Ferguson and Gao, 2018*). Among the various neocortical INT types, especially fast-spiking parvalbumin-positive interneurons (PVIs) have attracted substantial attention. Postmortem data indicated reduced numbers of PVIs and diminished expression of the GABA-synthetizing enzyme GAD in PVIs of the prefrontal cortex of schizophrenia patients (*Lewis et al., 2005*; *Volk et al., 2016*; *Marín, 2012*). Furthermore, eliminating prefrontal PVI output signaling induced working memory deficits in mice, underscoring the central role of PVI activity in cognitive function (*Murray et al., 2015*).

Transgenic *disrupted-in-schizophrenia-1* (Disc1)-mutant mice, which resemble an ultra-rare mutation with high penetrance for schizophrenia and major depressive disorder (*Blackwood et al., 2001*), show cognitive deficits, thus rendering Disc1-mutant mice a prime candidate to investigate the circuit mechanisms of a psychiatric disease (*Koike et al., 2006*; *Kvajo et al., 2008*). Much research has focused on the medial prefrontal cortex (mPFC) as an important center for cognitive function (*Goldman-Rakic, 1995*; *Sigurdsson et al., 2010*; *Cho et al., 2015*; *Sauer et al., 2015*; *Kim et al., 2016b*; *Chini et al., 2020*; *Duvarci et al., 2018*; *Kaefer et al., 2020*), but the circuit alterations underlying cognitive impairment are not well understood. In vitro studies in Disc1-mutants have put forward evidence for altered GABAergic (*Sauer et al., 2015*) and glutamatergic signaling in the mPFC (*Crabtree et al., 2017*). Moreover, in mice haploinsufficient for DISC1 isoforms (*Seshadri et al., 2015*), prefrontal PVIs receive reduced excitatory drive from afferents originating in the medio-dorsal thalamus (*Delevich et al., 2020*), a major input to the mPFC involved in cognitive function (*Bolkan et al., 2017*; *Parnaudeau et al., 2013*; *Schmitt et al., 2017*). In addition, overexpressing *Disc1* carrying a mutation previously associated with schizophrenia-related behavioral impairments (*Clapcote et al., 2007*) selectively in cortical pyramidal cells (PYRs) results in decreased PV expression in PVIs, suggesting that impaired communication in local cortical circuits might contribute to *Disc1*-mediated INT dysfunction (*Borkowska et al., 2016*). However, studies investigating the activity profile and local synaptic communication among prefrontal circuit components of Disc1-mutant mice in vivo are lacking. Furthermore, previous work identified reduced firing rates of PVIs in the nucleus accumbens of mice carrying an N-terminal truncation in the DISC1 gene in vivo (*Zhou et al., 2021*), but whether impairments of PVIs might contribute to altered network function in the mPFC of Disc1-mutant mice is not known.

Here, we interrogated PVI function in the mPFC of Disc1-mutant mice. In awake animals, fast-spiking INTs and optogenetically identified PVIs showed reduced firing levels. Furthermore, the proportion of INTs phase-coupled to fast gamma (60–90 Hz) oscillations was markedly reduced. Using spike train cross-correlation, we demonstrate that the excitatory effect of local glutamatergic synaptic inputs is reduced in vivo, consistent with impaired excitability of PVIs. On the network level, these alterations in INT function affected the activation of cortical cell assemblies, which are thought to be the building blocks of cortical information processing (*Buzsáki, 2010*).

## Results

### Impaired interneuron activity in the mPFC of Disc1 mice

Disc1-mutant mice have been shown to express impaired execution of working memory (*Koike et al., 2006*; *Kvajo et al., 2008*). To assess the potential network alterations underlying cognitive impairment, we performed single-unit recordings from the mPFC of Disc1-mutant (n = 9) and control mice (n = 7) that were awake in their home cage and analyzed discharge rates of PYRs (n = 720 and 613 in Disc1-mutant and control, respectively) and INTs (n = 104 and 79 in Disc1 and control, respectively; *Figure 1*, *Figure 1—figure supplement 1*). Both neuron types were separated based on spike waveform kinetics (*Sirota et al., 2008*; *Figure 1a*, *Figure 1—figure supplement 2*).

We found a significant reduction in INT firing rates in Disc1-mutant mice (U = 43, p=0.017, n = 7 mice each with electrophysiologically identified INTs, Mann–Whitney *U*-test), while the firing rates of PYRs were unaltered (t = 0.545, p=0.620, n = 9 Disc1-mutant and 7 control mice, unpaired *t*-test; *Figure 1b and c*). Reduced firing rates of INTs were preserved when we based the analysis on INTs with low activity levels (lowest third of the firing rate distribution, *Figure 1—figure supplement 3*). Similarly, unaltered PYR firing rates between genotypes were observed when only PYRs with high firing rate were considered (highest third of the firing rate distribution, *Figure 1—figure supplement 3*).

Since neocortical INT types are highly diverse (*Gupta et al., 2000*; *Druckmann et al., 2013*), putative INTs identified by their spike shape might include a mixed set of GABAergic populations. To specifically assess the activity profile of identified INTs, we focused on PVIs by crossing Disc1-mutant animals with mice expressing Cre-recombinase under the control of the PV-promoter (*Figure 1d*, PV-Cre-Disc1-mutant). Stereotaxic infusion of adeno-associated viruses encoding channelrhodopsin-2 (ChR2) and the red fluorophore tdTomato (tdT) flanked by double-inverted loxP sites (AAV-Flex-ChR2-tdT) into the mPFC rendered PVIs sensitive to blue light and allowed optogenetic identification

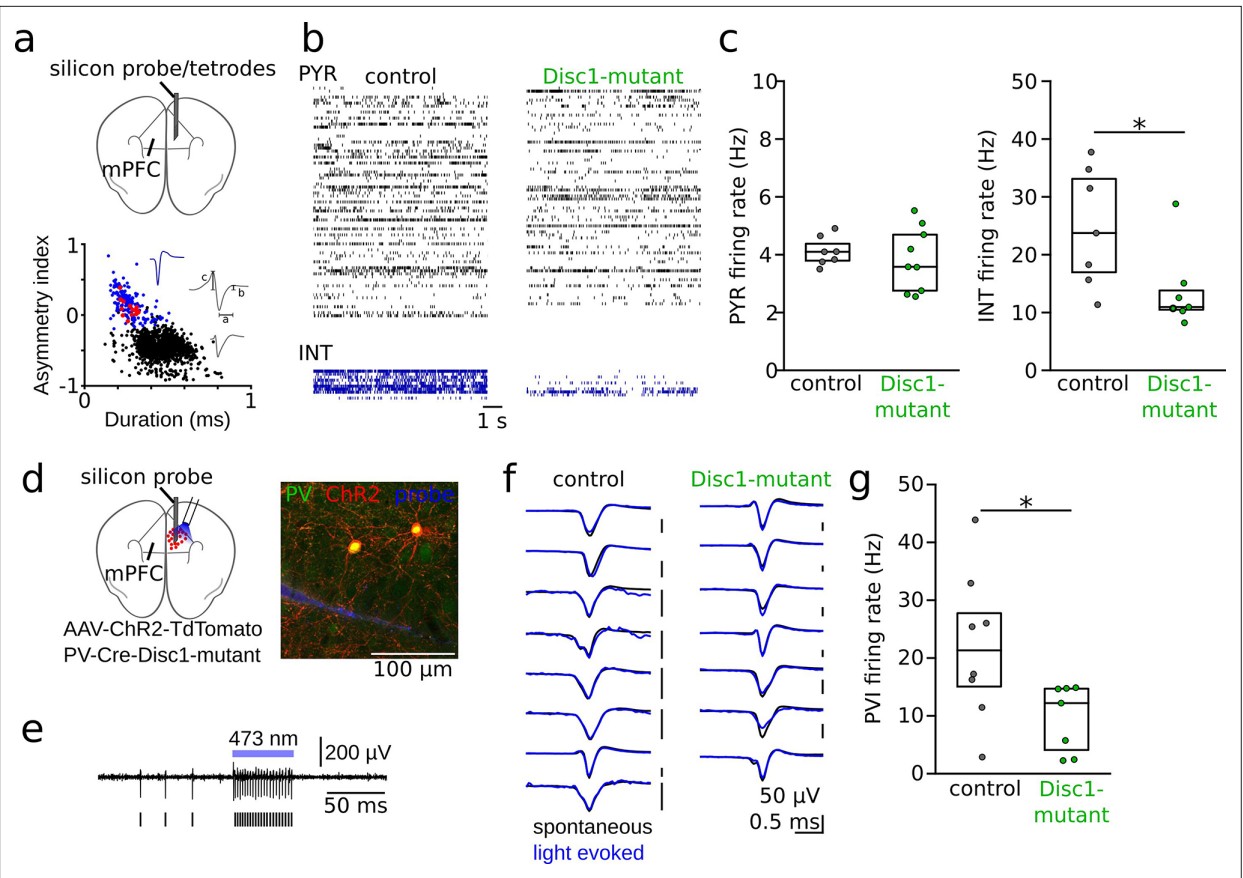

**Figure 1.** Reduced interneuron (INT) activity in the medial prefrontal cortex (mPFC) of Disc1-mutant mice. (**a**) Schematic of the recording configuration. Silicon probes or tetrode microdrives were targeted to the mPFC of freely moving mice. Bottom: identification of pyramidal cells (PYRs, black) and INTs (blue) based on duration (**a**) and asymmetry index (b – c)/(b + c) of filtered action potential waveforms. Red dots show optogenetically identified parvalbumin-positive interneurons (PVIs, see **d–g**). Blue and gray insets show an average unfiltered spike of an INT and PYR, respectively. (**b**) Example raster plots of PYR and INT activity in a freely moving control (black) and Disc1-mutant mouse (green). Each line shows the time series of spikes of a single neuron. (**c**) Decreased spike rates of INTs (Mann–Whitney *U*-test) but not PYRs (unpaired *t*-test) in the Disc1-mutant mPFC. Data points show averages of individual mice. (**d**) Schematic of the experimental strategy to label PVIs with ChR2. (**e**) Optogenetic identification of PVIs during blue laser illumination in awake head-fixed mice. Top: 0.3–6 kHz-filtered recording of one channel of the silicon probe (a single channel is shown for clarity). Bars indicate spike times of the unit. (**f**) Comparable average waveforms of light-triggered and spontaneous spikes of PVIs in both control (left) and Disc1-mutant mice (right). (**g**) Reduced spontaneous spiking of optogenetically identified PVIs in head-fixed PV-Cre-Disc1-mutant animals (Mann–Whitney *U*-test). Data points are individual PVIs. Boxes show median and upper/lower quartiles of the data distribution, *p<0.05.

The online version of this article includes the following figure supplement(s) for figure 1:

**Figure supplement 1.** Histological verification of recording sites in the medial prefrontal cortex (mPFC).

**Figure supplement 2.** Identification of pyramidal cells (PYR), interneuron (INT), and parvalbumin-positive interneurons (PVIs) in Disc1-mutant and control mice.

**Figure supplement 3.** Preserved interneuron (INT) but not pyramidal cell (PYR) rate differences in data subsampled by firing rate.

**Figure supplement 4.** Specific expression of ChR2 in parvalbumin-positive interneurons (PVIs) and comparable running speeds during head fixation.

**Figure supplement 5.** Kinetics of firing rate increase of parvalbumin-positive interneurons (PVIs) upon laser onset.

**Figure supplement 6.** Reduced firing rate of optogenetically identified parvalbumin-positive interneurons (PVIs) during UP-states under ketamine anesthesia.

of PV-expressing cells by brief laser pulses during single-unit recording (~10 mW, 50 ms; *Figure 1e and f*). Optogenetic identification experiments were performed in awake, head-fixed mice, running on a circular track ('Methods'). ChR2 expression was highly specific for PVIs (*Figure 1—figure supplement 4*). Moreover, spike waveform properties of PVIs clustered with electrophysiologically identified INTs (*Figure 1a*, *Figure 1—figure supplement 2*), and spike rates of light-sensitive cells increased rapidly upon laser onset (*Figure 1—figure supplement 5*), further supporting that our optogenetic

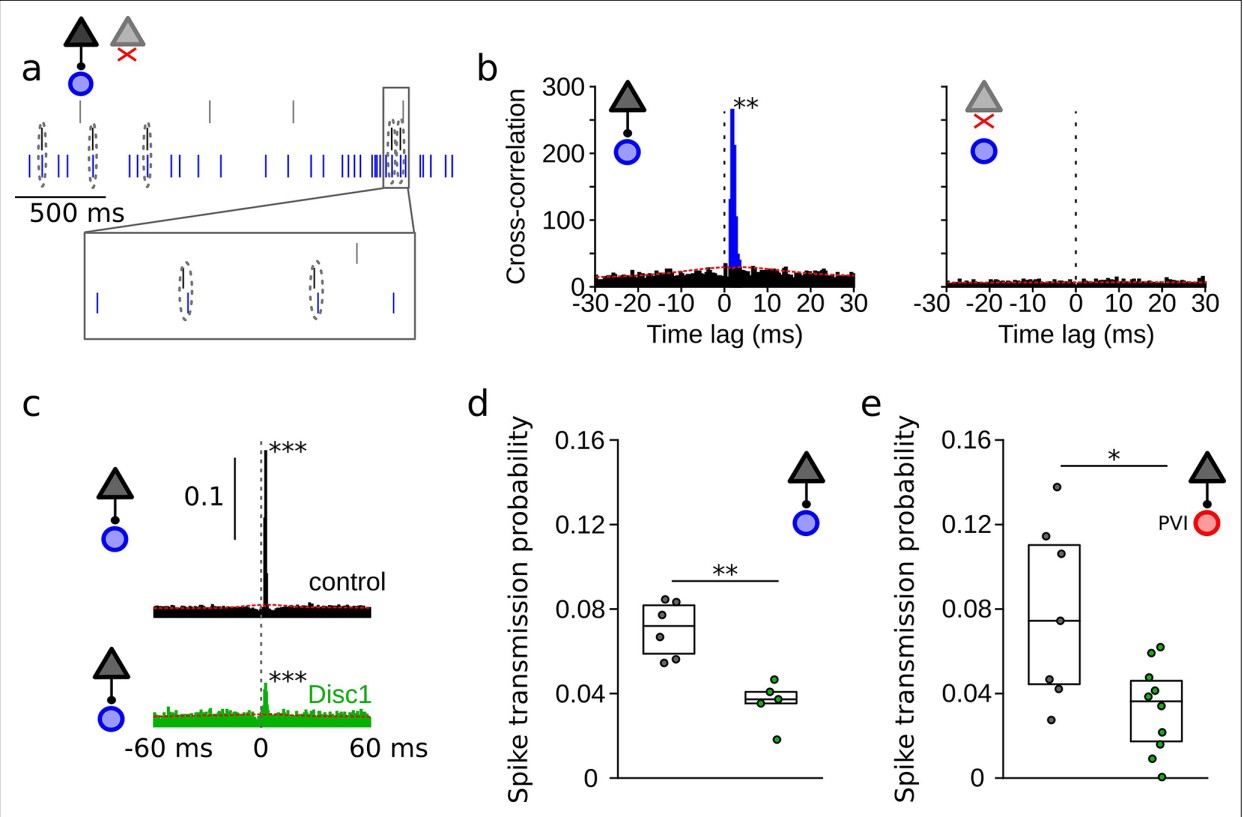

**Figure 2.** Reduced spike transmission at pyramidal cell–interneuron (PYR-INT) connections in the Disc1-mutant medial prefrontal cortex (mPFC).
(**a**) Short segment of spike trains of a PYR with (black) and without (gray) monosynaptic excitatory connection onto a GABAergic fast-spiking INT (blue).
Note that spiking in the black PYR consistently occurs briefly before an action potential in the INT. (**b**) Cross-correlograms of the units shown in (**a**) reveal
a sharp peak at 1–2 ms latency for the connected pair (left) but not for a non-connected pair of cells (right). Blue region shows short-latency spikes
significantly exceeding the slowly co-modulated baseline (red dotted line, see 'Methods'). (**c**) Example spike transmission probability histograms for
a connected PYR-INT pair in control (black) and Disc1-mutant mPFC (green). A significant synaptic connection was detected in both cases (p<0.001).
(**d**) Summary of spike transmission probabilities of all PYR-INT pairs in freely moving mice (unpaired ttest). Circles show mouse averages (**e**) Summary of
spike transmission probabilities of identified PYR-PVI connections in head-fixed mice (Welch's test). Circles represent individual connections. Boxes show
median and upper/lower quartiles of the data distribution, *p<0.05, **p<0.01.

identification protocol extracted ChR2-expressing PVIs. In the absence of light stimulation, the
discharge rate of PVIs was significantly reduced in PV-Cre-Disc1-mutant mice (U = 47, p=0.029,
Mann–Whitney *U*-test, n = 7 Disc1-mutant and 8 control cells from three mice each, *Figure 1g*). The
reduction in PVI activity in Disc1-mutant mice could not be explained by an altered speed modulation
because Disc1-mutant mice ran at comparable velocity as control mice (*Figure 1—figure supplement
4*). Furthermore, similar results of reduced PVI firing rates were obtained during recordings under
ketamine anesthesia in a separate cohort of mice (*Figure 1—figure supplement 6*). Thus, loss of
Disc1 expression results in impaired activity levels of INTs, including PVIs.

## Impaired spike transmission at PYR-INT synapses of Disc1-mutant mice

To assess whether reduced glutamatergic drive from local PYR might contribute to lower INT activity
levels, we investigated synaptic interactions from PYRs onto INTs in vivo (*Figure 2*). These connec-
tions can be captured by spike train cross-correlations in simultaneous recordings of large neuronal
populations (*English et al., 2017*; *Figure 2a–c*). PYR–INT connections were assessed by determining
spike transmission probability from baseline-corrected cross-correlograms of PYR and INT spike trains
(*English et al., 2017*; *Figure 2b*). Monosynaptic excitatory connections were apparent as significant
and sharp positive peaks in causal direction (i.e., spiking in the PYR preceding spiking in the INT
at short latency, *Figure 2b and c*). Among 7744 PYR-INT pairs tested, we detected 135 significant
monosynaptic interactions. The probability of finding PYR-INT connections was not different among

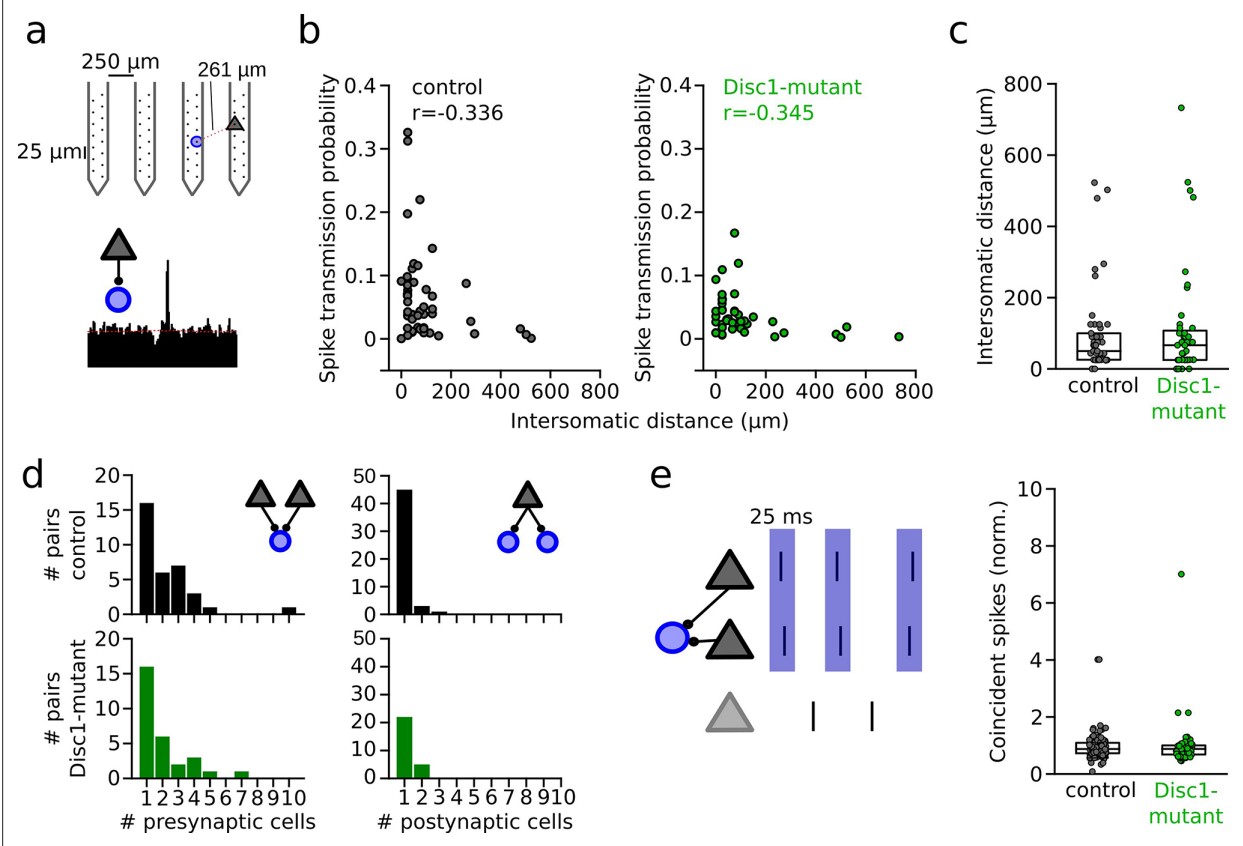

**Figure 3.** Unaltered network structure and presynaptic cooperativity of pyramidal cell–interneuron (PYR-INT) connections. (**a**) Example of distance calculation and corresponding cross-correlogram for a connected pair. (**b**) Summary of spike transmission probability as a function of intersomatic distance. Spike transmission inversely depends on intersomatic distance in both control (black) and Disc1-mutant pairs (green) (Spearman's correlation coefficients). Circles are individual PYR-INT pairs. (**c**) Summary of intersomatic distances of all PYR-INT pairs (Welch's test). (**d**) Quantification of the number of presynaptic (left) and postsynaptic partners (right) revealed comparable convergence and divergence, respectively (Mann–Whitney *U*-tests). (**e**) Left: schematic of synchronization of PYRs impinging on a shared postsynaptic INT. Right: summary of coincident spiking between convergent PYRs within 25 ms (Welch's test). Circles in (**b–e**) represent individual cells/cell pairs. Boxes show median and upper/lower quartiles of the data distribution.

The online version of this article includes the following figure supplement(s) for figure 3:

**Figure supplement 1.** Increased resonance boosting at pyramidal cell (PYR)-fast-spiking interneuron (INT) connections in Disc1-mutant mice.

genotypes (U = 36, p=0.158, n = 7 mice each, Mann–Whitney *U*-test). Cross-correlation analysis revealed significantly reduced spike transmission to ~51% of control levels in the mPFC of Disc1-mutant mice (t = 4.716, p=0.002, n = 5 Disc1-mutant and 6 control mice, unpaired *t*-test; *Figure 2c and d*). Furthermore, spike transmission at connections to optogenetically identified PVIs was reduced to ~42% (t = 2.653, p=0.029, Welch's test, n = 10 Disc1-mutant and 7 control connections, *Figure 2e*).

We performed a series of analyses to decipher whether reduced spike transmission reflects a proper reduction in the excitatory drive to local INTs or whether it might (in part) be explained by altered network properties. First, we assessed the somatic distance between presynaptic PYRs and postsynaptic INTs in a subset of mice, in which recordings were performed with silicon probes (n = 3 Disc1-mutant and 4 control mice, *Figure 3a*). In these recordings, the position of the pre- and postsynaptic neurons could be estimated due to the known physical location of the electrodes with the largest spike waveform deflections. Consistent with previous reports (*Zhou et al., 2021*), the strength of the spike transmission was inversely correlated with distance in both control (Spearman's *r* = −0.336, p=0.018, n = 49 connections) and Disc1-mutant connections (Spearman's *r* = −0.345, p=0.024, n = 43 connections, *Figure 3b*). Connected pairs in Disc1-mutant and control mice were similarly spaced (t = −0.458, p=0.648, n = 49 Disc1-mutant and 43 control connections, Welch's test, *Figure 3c*). These results rule out changes in distance-dependent signaling properties or systematic differences in inter-somatic distances of the recorded PYR-INT pairs as potential explanations of reduced spike

transmission. We moreover consistently observed converging input of multiple (>1) PYRs onto one INT with no significant difference in the number of presynaptic partners between genotypes (U = 536, p=0.527, n = 39 Disc1-mutant and 34 control connections, Mann–Whitney *U*-test, *Figure 3d*, left). Similarly, PYRs of both groups made frequent contact with more than one postsynaptic INT, with no significant difference in the number of divergent targets (U = 595.5, p=0.204, n = 27 Disc1-mutant and 49 control connections, Mann–Whitney *U*-test, *Figure 3d*, right), indicating intact overall PYR-INT network structure.

Second, synchronous presynaptic activity of PYRs with a common postsynaptic target has been demonstrated to boost spike transmission in the hippocampus (*English et al., 2017*). We extracted the pairwise synchronization of presynaptic PYRs with a common postsynaptic INT. Excess synchronization of convergent PYRs in a 25 ms window was not significantly different between Disc1-mutant and control mice (t = 0.280, n = 121 and 55 PYR pairs, p=0.780, Welch's test, *Figure 3e*). These findings rule out defective synchronization of PYRs as a mechanism of reduced spike transmission.

Third, hippocampal INTs show resonance membrane properties that boost spike transmission when postsynaptic spikes occur within an ~20–50 ms time interval (*English et al., 2017*). Consistent with reports from the hippocampus, neocortical PYR-INT connections displayed an approximately three-fold boost of spike transmission probability in the resonance time window of 20–50 ms, similar to the gain observed in CA1 (*English et al., 2017*; *Figure 3—figure supplement 1*). Moreover, identified PVIs showed approximately twofold resonance boosting of spike transmission (*Figure 3—figure*

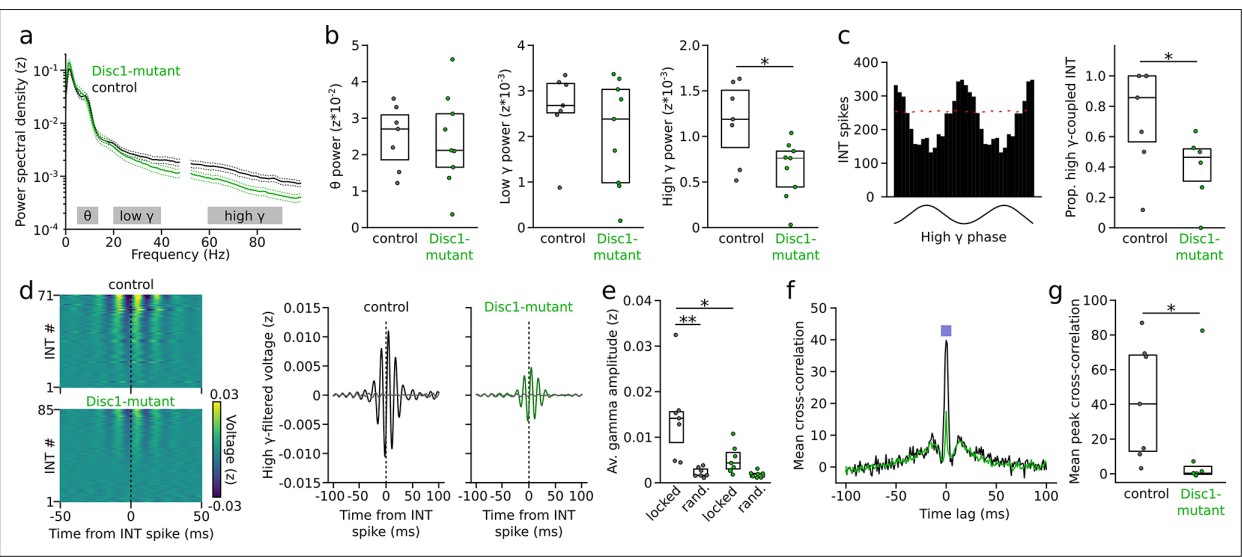

**Figure 4.** Impaired gamma oscillations and gamma phase-coupling of interneurons (INTs) in Disc1-mutant mice. (**a**) Average power spectral density of z-scored prefrontal local field potentials (LFPs) in control (black) and Disc1-mutant mice (green). Power at 50 Hz is omitted. Gray areas: Theta (6–12 Hz), low-gamma (20–40 Hz), and high-gamma bands (60–90 Hz) chosen for quantification of spectral power. (**b**) Average power in the theta, low-gamma, and high-gamma band. Circles are averages of individual mice (unpaired *t*-tests). (**c**) Left: example histogram showing the spike number of an INT as a function of high-gamma phase (shown on the bottom). Red line indicates the 95th percentile of the random distribution obtained by shuffling the interspike intervals (500 iterations). Crossing of the 95th percentile was used to determine significant phase-coupling. Right: reduced proportion of INTs with significant phase-coupling to high gamma in Disc1-mutant mice (unpaired *t*-test). (**d**) Left: average high-gamma-filtered LFP triggered by INT spiking (analyzed for all INTs with >1000 spikes). The data are sorted by the magnitude of the LFP deflection at the INT spike time. Right: grand averages of high-gamma oscillations aligned with INT spiking (solid lines) or at random time points (dashed lines). (**e**) Reduced INT spiking-triggered high-gamma amplitude in Disc1-mutant mice. Gamma amplitude triggered by INT spiking (locked) exceeded amplitude at random time points (rand.) in control but not Disc1-mutant mice (one-way ANOVA followed by Tukey tests). (**f**) Cross-correlation of the spike trains of simultaneously recorded INTs. The graph shows the grand average over all INTs (n = 432 and 215 pairs in Disc1-mutant and control mice, respectively). The inset illustrates the time window for the quantification of INT synchronization. (**g**) Reduced INT synchronization in a 10 ms time window in Disc1-mutant mice (Mann–Whitney *U*-test). Circles in (**b**, **c**, **e**, **g**) show averages of individual mice. Boxes show median and upper/lower quartiles of the data distribution, *p<0.05, **p<0.01.

The online version of this article includes the following figure supplement(s) for figure 4:

**Figure supplement 1.** Phase-coupling of pyramidal cells (PYRs) and interneurons (INTs) to gamma oscillations.

**Figure supplement 2.** Spike-triggered local field potential (LFP) gamma oscillations during interneuron (INT) spiking following presynaptic pyramidal cell (PYR) activity.

*supplement 1d*). However, the magnitude of the resonance gain reached similar levels for PYR-INT pairs in Disc1-mutant mice (t = 0.875, p=0.436, n = 53 Disc1-mutant and 71 control pairs, Welch's test, *Figure 3—figure supplement 1b*). In identified PVIs, resonance boosting of spike transmission was even significantly higher in Disc1 compared to controls (U = 11, p=0.022, n = 10 Disc1-mutant and 6 control connections, Mann–Whitney *U*-test, *Figure 3—figure supplement 1d*). Finally, reduced spike transmission in Disc1-mutant mice was also evident when we only considered spikes in the resonance time window (t = 4.530, p=0.0014, n = 5 Disc1-mutant and 6 control mice, unpaired *t*-test, *Figure 3— figure supplement 1c*). Jointly, these data suggest that while INTs in the Disc1-mutant mPFC display intact convergence, responses to synchronous input and resonance properties, they respond less to the action of local glutamatergic synaptic transmission.

## Defective gamma oscillations and INT synchrony in the Disc1 mPFC

The timed activation of PVIs is a crucial determinant of the emergence and synchrony of gamma oscillations (*Bartos et al., 2007*; *Sohal et al., 2009*; *Cardin et al., 2009*; *Perrenoud et al., 2016*). We therefore tested whether, in addition to firing rate deficits, oscillatory power and the temporal structure of INT spiking might be altered in Disc1-mutant mice. Power spectral density of local field potentials (LFPs) in the high-gamma frequency range (60–90 Hz) was significantly reduced to ~55% in Disc1-mutant mice (t = −2.728, p=0.016, n = 9 Disc1-mutant and 7 control mice, unpaired *t*-test, *Figure 4a and b*). In contrast, neither LFP power in the theta (6–12 Hz, t = −0.15, p=0.883) nor low-gamma regime (20–40 Hz, t = −1.033, p=0.319) were significantly affected (*Figure 4a and b*). Moreover, the proportion of INTs that discharged significantly phase-coupled to high-gamma oscillations was reduced from 73% in control to 39% in Disc1-mutant mice (t = −2.466, p=0.03, n = 6 Disc1-mutant and 6 control mice, unpaired *t*-test, *Figure 4c*) while no effect of genotype was observed for low-gamma oscillations (t = −0.897, p=0.393, n = 5 mice each, unpaired *t*-test, *Figure 4—figure supplement 1*). Preferred gamma phase and strength of gamma phase locking of significantly gamma-entrained INTs did not differ between genotypes (*Figure 4—figure supplement 1*). The proportion of significantly gamma-coupled PYRs was lower than that of INTs (~14% and ~25% in Disc1-mutant and control mice, respectively) and not significantly different between genotypes, irrespective of the gamma frequency band studied (low gamma: t = −2.62, p=0.8, n = 5 mice each; high gamma: t = −0.214, p=0.835, n = 5 Disc1-mutant and 6 control mice, unpaired *t*-tests; *Figure 4—figure supplement 1*).

To directly test whether reduced high-gamma power might be related to desynchronized INT spiking, we extracted high-gamma-filtered average LFPs triggered by spikes of INTs. This analysis revealed appreciable gamma oscillation power around the time of INT activity, which was not the case when random rather than INT spiking-related time windows were selected (*Figure 4d*). In agreement with previous work, these results suggest that INT spiking occurs synchronized with gamma oscillations. The amplitude of INT spike-triggered high-gamma oscillations was significantly reduced in Disc1-mutant mice compared to controls (t = −3.431, p=0.011, one-way ANOVA followed by Tukey tests, *Figure 4d and e*). Moreover, while spike-triggered gamma amplitudes significantly exceeded random-triggered amplitudes in control mice (t = 4.537, p=0.001), this effect was strongly reduced in Disc1-mutant mice and did not reach significance (t = 1.268, p=0.591, *Figure 4e*). Notably, restricting the analysis to INT spikes that followed spiking in a presynaptic PYR within 2 ms (i.e., likely reflecting activation of the INT by local PYRs) revealed no difference in the amplitude of spike-triggered gamma oscillations between genotypes (*Figure 4—figure supplement 2*). These results suggest that impaired gamma oscillations are not a simple reflection of defective excitatory drive in local PYR-INT loops. Consistent with the impaired high-gamma synchronization of INTs, we found reduced zero time-lag synchrony of pairs of simultaneously recorded INTs in Disc1-mutant mice (U = 42, p=0.026, n = 7 mice each, Mann–Whitney *U*-test, *Figure 4f and g*). These results jointly demonstrate impaired high-gamma oscillations and a decoupling of INTs from fast network oscillations in Disc1-mutant mice.

## Impaired neuronal assembly structure in the Disc1 mPFC

Previous experimental and simulation studies suggested that an important function of GABAergic inhibition might be to decorrelate synchronous activity of pyramidal cells imposed by shared inputs (*Renart et al., 2010*; *Tetzlaff et al., 2012*; *Sippy and Yuste, 2013*). Theoretical work further emphasized that feedback inhibition within a circuit is particularly effective at constraining pairwise spike train correlations (*Tetzlaff et al., 2012*). We therefore hypothesized that reduced inhibitory activity levels

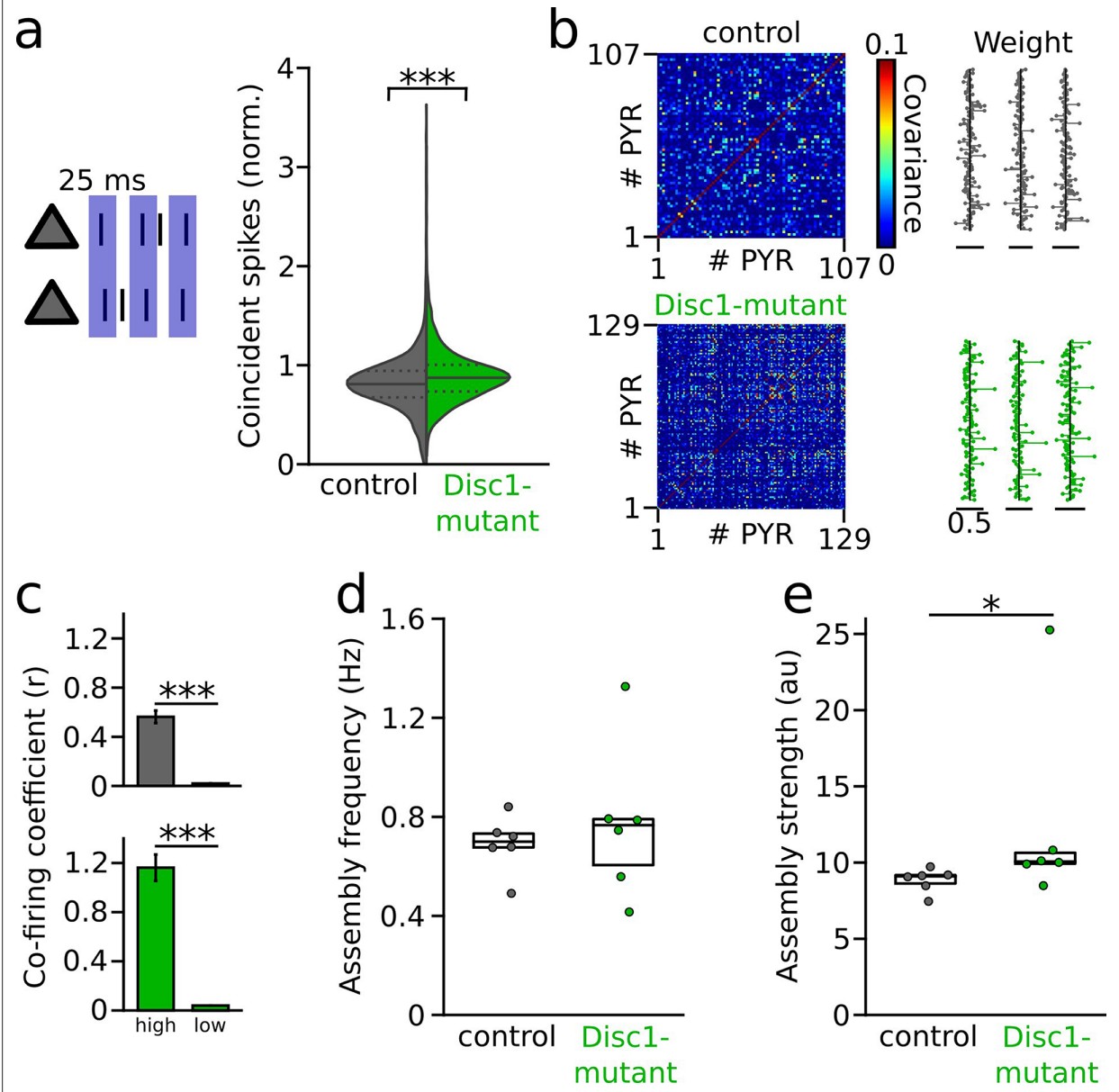

**Figure 5.** Enhanced synchronization and assembly activations in the Disc1-mutant medial prefrontal cortex (mPFC). (**a**) Left: schematic of synchronization of pyramidal cells (PYRs). Right: summary of coincident spiking within 25 ms (Welch's test). Solid line: mean, dashed lines: quartiles. (**b**) Detection of assembly patterns. Coactive patterns were extracted from spike covariance matrices (left) to retrieve assembly weight vectors (right). Data show examples from one control and Disc1-mutant mouse. (**c**) Neurons with high weight in an assembly pattern show stronger co-firing than neurons with low contribution (Welch's tests). (**d**) Assemblies activate on average at the same frequency in both genotypes (unpaired *t*-test). (**e**) Assembly activation strength is enhanced in Disc1-mutant mice (Mann–Whitney *U*-test). Circles in (**d**) and (**e**) show averages of individual mice. Boxes show median and upper/lower quartiles of the data distribution, *p<0.05, ***p<0.001.

might affect PYR coactivity in the Disc1-mutant mPFC. In the Disc1 mPFC, PYRs indeed displayed stronger pairwise synchronization at 25 ms time scale at the level of individual neuron pairs (t = −9.437, p=6.5 * 10$^{-21}$, n = 1797 control and 2445 Disc1 PYR pairs, this analysis considered all PYRs, *Figure 5a*). However, this analysis revealed no significant difference in the level of mouse averages (t = −0.178, p=0.861, n = 7 control and 9 Disc1-mutant mice, unpaired *t*-test).

Groups of coactive cells forming neuronal assemblies are thought to be a hallmark of cortical processing (*Buzsáki, 2010*; *El-Gaby et al., 2021*). To directly assess the expression of neuronal assembly patterns beyond pairwise comparisons, we retrieved coactive firing motifs using a step-wise

extraction procedure relying on principal and independent component analysis of binned spike trains of simultaneously recorded PYRs (*El-Gaby et al., 2021*; *van de Ven et al., 2016*; *Folschweiller and Sauer, 2022*; *Figure 5b*). Using this procedure, we detected in total 113 assembly patterns in control mice (n = 6) and 122 patterns in Disc1-mutant mice (n = 6), with one assembly pattern found per 5.3 ± 0.6 and 5.1 ± 0.8 PYRs in control and Disc1 mPFC, respectively (U = 19, p=0.936, n = 6 mice each, Mann–Whitney *U*-test). Neurons with large weight in a detected assembly pattern showed more pronounced co-firing than neurons that contributed less to a given pattern (*Figure 5c*), corroborating the validity of the assembly detection method. Spontaneous assembly patterns activated at comparable rates in control and Disc1-mutant mice (t = −0.595, p=0.565, six mice each, unpaired *t*-test, *Figure 5d*). However, consistent with enhanced pairwise co-firing detected on the level of individual cells, the assembly activation strength was significantly larger in Disc1-mutant mice (U = 4, p=0.026, n = 6 mice each, Mann–Whitney *U*-test, *Figure 5e*). These data jointly indicate altered assembly structure in Disc1-mutant mice.

## Discussion

Despite the broad recognition that fast-spiking INTs, particularly PVIs, contribute to neuronal network synchronization at gamma frequencies (*Bartos et al., 2007*; *Sohal et al., 2009*; *Cardin et al., 2009*), show task-dependent tuning during working memory (*Kim et al., 2016b*; *Lagler et al., 2016*), and are required for the proper execution of working memory (*Murray et al., 2015*; *Murray et al., 2011*), information is still lacking on PVIs' activity in the prefrontal cortex of working memory-deficient Disc1-mutant mice. A major consequence related to this gap of knowledge is our limited understanding of the underlying pathophysiology, which prevents the development of new therapies for mental diseases. Here, we provide a first in vivo electrophysiological characterization of fast-spiking INT and PVI activity in Disc1-mutant mice. Extending upon previous electrophysiological Disc1 studies (*Sauer et al., 2015*; *Chini et al., 2020*; *Delevich et al., 2020*; *Kaefer et al., 2019*), we show that the average discharge rates of INTs are reduced in vivo by ~44%, whereas PYR activity is unaltered (*Figure 1b and c*).

Related to the reduced mean PVI activity, we detected an ~41% reduction in the magnitude of spike transmission probability at PYR-PVI connections with unaltered probability of finding connected PYR-INT pairs in Disc1-mutant mice (*Figure 2d and e*). Although the firing rates and spike waveform properties of PVIs were indistinguishable from electrophysiologically identified putative fast-spiking INTs (*Figure 1—figure supplement 2*), it should be noted that a limitation of our study is the comparable low number of optogenetically identified PVIs. Spike transmission is an indirect estimation of synaptic connections onto INTs (*English et al., 2017*). Control analyses demonstrated that the weaker spike transmission is unlikely to be caused by network effects such as altered convergence or intersomatic distance between the connected cells, suggesting that it reflects a genuine reduction in excitatory drive at PYR-INT connections. Given the lower spike transmission probability in Disc1-mutant mice, it is, however, likely that the number of detected pairs might have been undersampled in Disc1-mutant mice. The cellular and network mechanisms underlying reduced spike transmission to INTs could not be determined in this study. Reduced numbers of excitatory connections from local PYRs onto INTs might compose one contributing factor. Indeed, Disc1-mutant mice show altered wiring of excitatory connections in the hippocampal dentate gyrus-to-CA3 pathway (*Kvajo et al., 2011*), although analysis of the mPFC revealed no apparent changes in excitatory axon distribution (*Crabtree et al., 2017*). Another potential factor might be the insufficient expression of *N*-methyl-ᴅ-aspartate receptors (NMDARs) in cortical INTs. Removing NMDARs specifically from cortical PVIs induces working memory and prepulse inhibition impairments in mice, reminiscent of symptoms observed in schizophrenia patients (*Belforte et al., 2010*). Furthermore, it is possible that the observed lower spike transmission probability is caused by reduced excitability of INTs in Disc1-mutant mice. For instance, blocking T-type calcium channels in PVIs results in reduced numbers of cfos-positive PVIs in a social interaction task (*Shen et al., 2021*). Notably, reduced expression of T-type channels has been reported in mice upon PVI-specific deletion of the schizophrenia-associated gene FXR1 (FMR1 autosome homolog 1), although potential defects in T-type channel expression specifically in Disc1-mutant mice are unclear (*Shen et al., 2021*). Although whole-cell recordings from PVIs in prefrontal slices of Disc1-mutant mice failed to detect changes in intrinsic excitability in vitro (*Delevich et al., 2020*), it cannot be ruled out that such differences might exist in the intact circuitry

under conditions of physiological neuromodulatory inputs. This is particularly relevant because PVIs in slice preparations of mice expressing dominant-negative DISC1 (*Hikida et al., 2007*) show reduced current injection-induced firing rate increases in response to bath application of dopamine 2 receptor agonists (*Cardarelli et al., 2018*).

On the level of neuronal networks, we propose that reduced drive to INTs induces a chain of interrelated functional consequences. Diminished recruitment of PVIs by local glutamatergic inputs will result in lower PVI firing rates and weaker PVI-mediated synaptic inhibition. Lower synchrony of PVIs might consequently result in the observed reduction in gamma power in Disc1-mutant mice (*Kim et al., 2016a*; *Wang and Carlén, 2012*). Moreover, a gamma oscillation dysfunction in the mPFC might contribute to working memory deficits in Disc1-mutants. This proposal fits to reports of high-gamma activities occurring during the execution of working memory in mice (*Yamamoto et al., 2014*) and monkeys (*Lundqvist et al., 2016*).

One might expect that reduced PVI activity and consequently reduced perisomatic inhibition (*Courtin et al., 2014*) may lead to enhanced PYR discharge rates. However, we did not observe significant changes in the average activity or gamma phase relationship of PYRs. Since perisomatic inhibition generates large GABA$_A$ receptor-mediated conductances at unitary GABAergic connections (*Jouhanneau et al., 2018*), the compound inhibitory conductance emerging from the convergence of active GABAergic perisomatic inputs onto a single PYR might be sufficiently large to control the activity of PYR populations. This hypothesis fits to previous studies showing that pharmacogenetic silencing of hippocampal PVIs or the elimination of PVI synaptic output with tetanus toxin had no effect on the expression of the activity marker cfos (*Murray et al., 2011*; *Stefanelli et al., 2016*). Additional factors might contribute to preserved PYR firing rates. It is conceivable that synaptic connections among local PYRs might be reduced in their efficacy. However, due to the small amplitude of excitatory postsynaptic potentials at PYR-PYR connections it is challenging to detect them in vivo for the quantification of spike transmission probability (*Campagnola et al., 2022*). Moreover, it remains unclear whether firing rates of other neocortical interneuron types such as somatostatin-positive INTs are altered in Disc1-mutant mice. Quantifications of cfos expression levels revealed that somatostatin-positive INTs efficiently control activity levels of local principal cells in the hippocampus (*Stefanelli et al., 2016*). Enhanced inhibitory tone provided by these INTs might, thus, counteract the reduced inhibitory inputs from PVIs.

In addition to effects on firing rates, it has been argued that inhibition might counteract excessive synchronization of PYRs by effectively decorrelating shared synaptic inputs (*Renart et al., 2010*; *Tetzlaff et al., 2012*; *Sippy and Yuste, 2013*). In line with this hypothesis, chemogenetic silencing of PVIs enhanced pairwise correlations among cortical PYRs (*Hamm et al., 2017*). Supporting the synchronization-limiting role of feedback inhibition, we found increased spike correlations among cortical PYRs at the level of individual cell pairs in Disc1-mutant mice. This observation was not restricted to pairwise synchronization but generalized to assemblies of coactive neurons, which showed stronger expression strengths. The expression strength of neuronal assemblies as evaluated with the method chosen here reflects the number of coactive cells, their synchronization and firing rate. Importantly, we chose a time window of 25 ms to detect assembly patterns, which is within the integration time of cortical PYRs (*Koch et al., 1996*). Active assemblies within this integration window are, thus, expected to efficiently affect downstream reader neurons (*Buzsáki, 2010*). Disturbed cell assembly expressions have been identified in the chronic ketamine and Df(16 )A$^{+/-}$ models of schizophrenia (*Hamm et al., 2017*). Notably, in those models, stability of assembly patterns rather than their expression strength were affected. These data imply that while the nature of the assembly dysfunction might differ between distinct models, altered assembly expressions might be a common final path underlying cognitive disturbances in schizophrenia.

What are the potential functional implications of increased assembly activation strength for cognition? Enhanced activation strength means that neurons that are part of an assembly tend to co-fire more strongly in Disc1-mutant mice. Recurrent excitatory connections and Hebbian plasticity among PYRs are thought to underlie the formation and maintenance of cell assemblies (*Harris, 2005*; *Palm et al., 2014*). Previous work identified increased synaptic plasticity at layer 2/3-to-layer 5 glutamatergic synapses of Disc1-mutant mice (*Crabtree et al., 2017*), which might support the emergence of stronger assemblies in addition to reduced inhibitory input. Given the stronger reactivation of cell assemblies in the mPFC of Disc1-mutant mice, we propose they might retain larger stability over time.

This hypothesis could be tested by calcium imaging of population activity over time (*Hainmueller and Bartos, 2018*). In the visual cortex, both stable (i.e., repeatedly active over weeks) and transiently active (i.e., largely restricted to one experimental session) assemblies have been identified using in vivo calcium imaging methods (*Pérez-Ortega et al., 2021*). It has been previously suggested that stable cell assemblies might relate to representations of long-term memories, while transient cell assemblies could reflect the emergence of new assemblies (*Pérez-Ortega et al., 2021*). This conjecture proposes a trade-off between stable representations and dynamic encoding of memory content. It is possible that this trade-off is shifted toward the stable side in Disc1-mutant mice, which might possibly hinder the flexible allocation of new compared to previously encoded memory contents. Since mPFC-dependent working memory requires dynamic updating of memory, a shift toward stable representation might contribute to impaired working memory performance of Disc1-mutant mice (*Kvajo et al., 2008*).

## Methods

### Animals

Disc1-mutant (*Koike et al., 2006*), PV-Cre (RRID:IMSR_JAX:008069), PV-Cre;Disc1-mutant and wild-type littermate control mice were maintained on a 12 hr dark-light cycle with free access to food and water. Disc1-mutant mice were a gift from J. Gogos. At the start of the experiment, animals were at least 6 weeks old. Mice of both sexes were used in this study. All experiments were approved by the Ethics Committee of the Regierungspräsidium Freiburg (license numbers G16-152 and G19-22).

### Microdrive implantation and electrophysiological recording

Mice were anesthetized with isoflurane in oxygen (induction: 3%; maintenance: 1–2%) and placed on a heating pad. Analgesia was achieved by subcutaneous injection of buprenorphine (0.05–0.1 mg/kg body weight). A four-shank silicon probe (model P-1, Cambridge NeuroTech) mounted on a microdrive (NanoDrive, Cambridge NeuroTech) was implanted in rostro-caudal orientation with the most anterior shank positioned at ~2 mm anterior from bregma and 0.35–0.4 mm lateral to the midline. A pair of stainless-steel screws (M1) was inserted into the bone over the cerebellum and connected to the ground and reference leads of the electrode interface board. Additional screws were inserted into the parietal and contralateral frontal bone for further stability. In a subset of mice, microdrives containing eight tetrodes build from 12.5 µm tungsten wire were targeted to the mPFC using the following coordinates: 1.8–2 mm anterior of bregma, 0.45 mm lateral of the midline, and 1.8–2.8 mm below bregma. The microdrive was fixed to the skull with dental cement (SuperBond). After surgery, analgesia was provided for 2 days with buprenorphine (subcutaneous injections every 6 hr during daytime and in the drinking water overnight) and carprofen (4–5 mg/kg body weight, subcutaneous injection every 24 hr for 2 days). About 1 week after surgery, wide-band neural signals (0.1 Hz to 7.5 kHz, 32–64 channels) were recorded with tethered RHD2000 amplifiers (Intan Technologies) at 30 kHz sampling frequency while animals were awake in their home cage. Data acquisition was performed with the OpenEphys GUI (*Siegle et al., 2017*). After the recording, the microdrive was lowered by 100–200 µm to obtain access to a new set of units on the next recording day. Up to five recording sessions were performed with each animal. After completion of the recordings, the mice were deeply anesthetized by an intraperitoneal injection of urethane (2 g/kg body weight) and transcardially perfused with ice-cold phosphate-buffered saline (~10 ml) followed by freshly prepared ice-cold 4% paraformaldehyde (50–100 ml). In a subset of animals, electrolytic lesions of the recording sites were done with DC stimulation (10–20 V) applied to a subset of electrodes. After post-fixation overnight, 100 µm sections were cut with a vibratome (Leica VT1200 S) and stained with DAPI. Epifluorescence microscopy was used to assess the location of the recording sites. Recordings from the prelimbic, cingulate, and infralimbic cortex were included and pooled for analysis.

### Viral injections

Animals were anesthetized with isoflurane as described above. AAV1-flex-ChR2-Tdtomato (Addgene, titer 2 * $10^{12}$/ml, 0.5 µl) or AAV1-flex-ChR2-mCherry (Charite Vector core, titer 4 * $10^{11}$/ml, 0.5 µl) was slowly infused into the right or left mPFC at two anterioposterior locations (1.5 and 2 mm anterior from bregma) using thin glass pipettes fabricated with a microfilament puller (Flaming Brown). After

viral injection, the pipette was kept in place for 5 min to ensure efficient virus diffusion into the tissue. Acute recordings commenced >2 weeks after virus injection.

## Acute recordings

For recordings under anesthesia, the mice were anesthetized with isoflurane and a craniotomy (~2 mm wide) was performed above the right mPFC. After removal of the dura, the craniotomy was sealed off with dura gel (Cambridge NeuroTech) and a custom 3D-printed head bar with a circular ring around the craniotomy site was fixed to the skull with dental cement. The animal was then immediately transferred to the recording station while anesthesia was maintained throughout the recording session with intraperitoneal injections of ketamine and xylazine (initial dose: 100 and 13 mg/kg body weight, respectively, topped up by 10–20% every 20–40 min). For awake recordings, a steel head plate was implanted on the skull and the animals were allowed to recover from head plate implantation for 3 days. For habituation to head fixation, the mice were briefly sedated with isoflurane and head-fixed such that they could comfortably stand on a circular Styrofoam weal. A virtual reality (circular track, length 1.5 m, visual cues placed outside the arena) was constructed with open-source 3D rendering software (Blender) and was projected on five computer screens surrounding the head-fixation setup (*Schmidt-Hieber and Häusser, 2013*). Over subsequent days, mice were accustomed to head fixation by daily increasing the time of head fixation until the animals appeared calm and traversed the circular maze reliably. Once the animals were habituated, a craniotomy was performed as described above. Carprofen was injected subcutaneously on the day of the surgery. A four-shank silicon probe (Cambridge NeuroTech) coated with fluorescent marker (DiD) was slowly (~5–10 μm/s) lowered to the mPFC (935–1758 μm below brain surface). Wide-band neural signals were recorded at 30 kHz sampling with a 64-channel amplifier (Intan Technologies) connected to a USB acquisition board (OpenEphys; *Siegle et al., 2017*). Laser light (473 nm, ~10 mW intensity at the fiber tip, 50 ms pulses at 0.1 Hz) was delivered through a 200 μm optical fiber glued to the silicon probe. Afterward, the silicon probe was slowly retracted and the animals were transcardially perfused with ~20 ml phosphate-buffered saline followed by ~30 ml of 4% paraformaldehyde. After post-fixation overnight, 100-μm-thick frontal sections of the mPFC were cut and stained with rabbit-anti-PV antibody (1:1000, Swant PV27) and DAPI. The location of the silicon probe and the immunostaining were visualized with a laser-scanning microscope (LSM 710, Zeiss). Recordings in the prelimbic and cingulate cortex were pooled for analysis.

## Single-unit isolation

Single unit clusters were isolated from bandpass-filtered raw data (0.3–6 kHz) using MountainSort (*Chung et al., 2017*). Putative single-unit clusters that fulfilled quality criteria of high isolation index (>0.90) and low noise overlap (<0.1) were kept for manual curation, during which only clusters with a clear refractory period and clean waveform shape were saved for further analysis. In case of two clusters with similar waveforms, cross-correlation was used to assess whether clusters had to be merged. Individual tetrodes or shanks of a silicon probe were clustered separately. Isolated units were separated into excitatory and inhibitory neurons based on trough-to-peak duration and asymmetry index (*Sirota et al., 2008*) using *k*-means clustering. To analyze physical distances between units, the channel on the silicon probe with the largest negative amplitude deflection was defined as the location of the unit and the absolute inter-somatic distance was calculated.

## Optogenetic tagging

To detect ChR2-expressing PVIs, we applied 50 ms pulses of blue laser light. The firing rate during the last 25 ms of the pulse was calculated and averaged for all pulses. Next, we created a random spike train of the unit by shuffling the interspike intervals and calculated the average spike frequency during the second half of the light pulse. The shuffling was repeated independently 1000 times. A unit was considered significantly light-sensitive if the actual rate during light delivery exceeded the 99th percentile of that of the shuffled distribution. To assess the firing rate change of identified PVIs relative to laser onset in *Figure 1—figure supplement 5*, we convolved the spike trains of PVIs with a Gaussian kernel (SD = 5 ms) and extracted the averaged firing rate change triggered by laser onset.

## Spectral analysis and spike-field coupling

For power spectral density analysis, the raw traces were downsampled to 1 kHz and z-scored. Power was measured with no additional normalization in a sliding window (length: 1 s) with fast Fourier transforms (zero-padding: 10 s, using the SciPy function *welch*). Power was averaged across five randomly chosen channels for all sessions for each mouse.

To assess phase-coupling to the LFP, the raw LFP trace was downsampled to 1 kHz and filtered in the low- (20–40 Hz) or high-gamma range (60–90 Hz). The instantaneous amplitude of the signal was computed using Hilbert transformation, and the time points at which the gamma amplitude was larger than the $80^{th}$ percentile were identified. Phase-coupling analysis was restricted to these time points. The instantaneous phase at each time of spiking of a given neuron was extracted using the Hilbert transform. This array was used to obtain spike-gamma phase histograms (18 bins). Significant coupling was tested against surrogate data obtained by shuffling the interspike intervals and creating spike-phase histograms for each shuffled spike train (500 iterations). Significant phase-coupling was defined as at least two consecutive bins of the spike-phase histogram exceeding the $95^{th}$ percentile of the shuffled histograms. The coupling depth of significantly coupled cells was determined using the pairwise phase consistency (*Tamura et al., 2016*) defined as the average pairwise circular distance $D$:

$$D = \frac{2}{N(N-1)} \sum_{j=1}^{N-1} \sum_{k=(j+1)}^{N} d\left(\Theta_j, \Theta_k\right),$$

where d gives the absolute angular distance between the phase angles $\Theta$ of spikes *j* and *k* of a unit, and N is the total number of spikes. *D* is normalized to obtain pairwise phase consistency PPC as

$$PPC = \frac{\pi - 2D}{\pi}$$

To account for the effects of firing rate, 250 randomly selected spikes were used in 100 iterations for each neuron to estimate PPC from the average PPC value across iterations.

## Spike-triggered LFP analysis

The raw LFP of one channel in the vicinity of the neuron (i.e., on the same shank of a silicon probe or from within the same tetrode as the channel with largest amplitude of the unit) was z-scored and filtered in the high-gamma range (60–90 Hz). Spike-triggered filtered LFPs were extracted in a window of 500 ms before and after each spike and averaged for each INT with at least 1000 spikes. Amplitude of the average signal was obtained by the absolute of the Hilbert transform measured in a time window of 20 ms around the INT spike time. As a control, the same extraction procedure was applied to a matching number of randomly chosen time points in the recording. In addition, we extracted INT spikes that followed activity in a presynaptic PYR within 2 ms and performed the same spike-triggered gamma amplitude analysis as before.

## Analysis of INT synchronization

Spike trains of simultaneously recorded INT were cross-correlated using *neuronpy.filter_correlogram* without filter kernel at 100 ms maximal lag. Synchronization was measured by taking the mean cross-correlation in a 10 ms window around zero time lag, normalized by subtracting the mean of the cross-correlation at –100 to –50ms time lag.

## Spike transmission analysis

To detect monosynaptic excitatory interactions between PYRs and INTs, we utilized cross-correlation methods (*English et al., 2017*). Cross-correlations between spike trains (0.4 ms bins, maximal time lag: 50 ms) were calculated if both units fired at least 500 spikes during the recording interval. Criteria for a significant monosynaptic interaction were a peak in the monosynaptic time window (0.8–2.8 ms following the spike in the PYR) significantly exceeding the co-modulated baseline and the peak in anti-causal direction (i.e., INT-PYR, –2 to 0 ms). The baseline *b* was obtained by convolving the raw cross-correlogram with a partially hollowed Gaussian function (hollow fraction: 0.6; standard deviation: 10 ms). The Poisson distribution with continuity correction was used to estimate the probability of the observed magnitude of cross-correlation in the monosynaptic bins ($P_{syn}$):

$$P_{syn} = 1 - \sum_{x=0}^{n-1} \left( \frac{e^{-b(m)} b(m)^x}{x!} \right) - 0.5 \frac{e^{-b(m)} b(m)^n}{n!}$$

Similarly, we estimated the probability of the observed count in the monosynaptic bins of the cross-correlogram being larger than the count in anticausal direction ($c_{anticausal}$) using the Poisson distribution with continuity correction:

$$P_{causal} = 1 - \sum_{x=0}^{n-1} \left( \frac{e^{-c_{anticausal}(-m)} c_{anticausal}(-m)^x}{x!} \right) - 0.5 \left( \frac{e^{-c_{anticausal}(-m)} c_{anticausal}(-m)^n}{n!} \right).$$

Following optogenetic ground truth data obtained in the hippocampus, a pair was marked as connected if $P_{syn} < 0.001$ and $P_{causal} < 0.0026$ (**English et al., 2017**). Spike transmission probability was defined as the spiking in the monosynaptic window exceeding $b$ normalized by the number of presynaptic spikes.

## Presynaptic synchronization and resonance analysis

To assess presynaptic synchronization of PYRs with shared postsynaptic partner, we quantified the proportion of spikes of the two neurons that co-occurred within a time window of 25 ms. The data were then normalized by the expected rate of co-occurrence by dividing by 2× frequency of comparison train × synchrony window × number of spikes in the reference train using the function *coincident_spikes* of the Python package *neuronpy.utils.spiketrain*.

To quantify resonance properties, presynaptic spikes that occurred at times when the postsynaptic neuron had fired 20–50 ms before the reference spike (resonance time window) and neither the pre- nor the postsynaptic unit had fired between the end of the resonance time window and the reference spike were isolated. Spike transmission probability was computed separately for these extracted resonance spikes and compared to the transmission probability obtained for all presynaptic spikes.

## Detection of cell assemblies

The spike trains of PYRs were binned in 25 ms bins and normalized by z-scoring. The number of assembly patterns was detected from the binned spike train matrix using the Marčenko–Pastur law (**El-Gaby et al., 2021**; **van de Ven et al., 2016**; **Folschweiller and Sauer, 2022**). The Marčenko–Pastur law states that a covariation matrix constructed from statistically independent random variables (such as neurons that do not co-fire) gives eigenvalues below a critical value (**Marčenko and Pastur, 1967**; **Lopes-dos-Santos et al., 2013**). If neuronal firing occurs correlated with each other (as would be the case for assemblies), eigenvalues above the critical limit should exist. The number of eigenvalues thus indicates the number of assembly patterns. We determined the eigenvalue limit $l$ as

$$l = \left( 1 - \sqrt{\frac{1}{q}} \right)^2$$

with

$$q = \frac{N_{bins}}{N_{neurons}}$$

Using the fastICA algorithm of scikit.learn (**Pedregosa et al., 2011**), we extracted the number of independent components given by the eigenvalues above $l$. The resulting components are the weight vectors of each assembly. Since the orientation of independent components is arbitrary, each vector was oriented to have the largest deflection in positive direction and was further scaled to unit length (**van de Ven et al., 2016**). Assembly neurons were defined as those cells with a weight exceeding 2× the standard deviation of the pattern vector. To assess co-firing of neurons with large weight, Pearson's $r$ was measured for the binned spike trains of those neurons and compared to the correlation values among all other PYRs. Sessions with at least 10 PYR were used for this analysis (Disc1-mutant: 18 sessions from five mice; control: 15 sessions from four mice).

## Reconstruction of assembly activations over time

To obtain the activation strength $A$ of assembly patterns, the weight vectors were projected on smoothed spike trains of all simultaneously recorded pyramidal neurons as

$$A_t = Z_t^T P Z_t$$

where $Z$ is the smoothed and z-scored spike train of each neuron obtained by convolving the spike trains with a Gaussian function, $P$ is the outer product of the respective weight pattern, and $T$ is the transpose operator. Significant pattern activations were defined as threshold crossings of 5 (*van de Ven et al., 2016*; *Folschweiller and Sauer, 2022*). Average activation strength of a given pattern was calculated for all data points above that threshold. To measure assembly reactivation, the pattern detection was applied to the first half of the recording. Then, the expression strength over time was separately computed for the first and second half of the recording. Reactivation strength was measured as the average above-threshold strength of each pattern during the second half divided by the strength during the first half.

## Statistical analysis

Comparisons of two groups were performed with a two-sided Welch's test if the number of observations was >30 in each group. For n < 30 in each group, data were compared by two-sided Student's or Welch's *t*-tests in case of comparable or different standard deviations, respectively, in case that data passed the normality test (Shapiro–Wilk test), otherwise a two-sided Mann–Whitney *U*-test was used. Paired comparisons were performed with paired *t*-test (two-tailed). Connection probabilities were compared with a $\chi^2$ test. Differences in the preferred phase of spiking relative to gamma oscillations were assessed with a two-sample permutation test. First, the circular distance between the circular means of both groups was measured. Second, the data of both groups were pooled and randomly redrawn into two groups, and the circular distance between the circular means of the redrawn groups was taken (10,000 iterations). Finally, a p-value was calculated by counting the number of times the actual circular distance was larger than the redrawn distance divided by the number of iterations. All analyses (except for initial spike sorting, see above) including statistics were performed using Python 2.7. Design and reporting of this study followed ARRIVE guidelines (except blinding [not done] and randomization [not applicable]).

## Acknowledgements

We thank Karin Winterhalter and Kerstin Semmler for technical assistance, Sebastian Kugler for contributing to pilot experiments, and Joseph Gogos for providing Disc1-mutant mice. This work was supported by the Deutsche Forschungsgemeinschaft (FOR2143-2, BA1582/2-2, MB; SA3609/1-1, J-FS; FOR5159 TP07, J-FS and MB), the European Research Council Advanced Grant (ERC-AdG 787450, MB), and the Excellence Initiative of the German Research Foundation (Brain-Links Brain-Tools, MB).

## Additional information

### Funding

| Funder | Grant reference number | Author |
| --- | --- | --- |
| Deutsche Forschungsgemeinschaft | SA3609/1-1 | Jonas-Frederic Sauer |
| Deutsche Forschungsgemeinschaft | BA1582/2-2 | Marlene Bartos |
| Deutsche Forschungsgemeinschaft | FOR2143-2 | Marlene Bartos |
| Deutsche Forschungsgemeinschaft | FOR5159 TP7 | Jonas-Frederic Sauer Marlene Bartos |
| European Research Council | ERC-AdG787450 | Marlene Bartos |

The funders had no role in study design, data collection and interpretation, or the decision to submit the work for publication.

## Author contributions
Jonas-Frederic Sauer, Conceptualization, Software, Formal analysis, Funding acquisition, Investigation, Visualization, Methodology, Writing – original draft; Marlene Bartos, Conceptualization, Supervision, Funding acquisition, Project administration, Writing – review and editing

## Author ORCIDs
Jonas-Frederic Sauer http://orcid.org/0000-0002-6854-7294
Marlene Bartos http://orcid.org/0000-0001-9741-1946

## Ethics
All experiments were approved by the Ethics Committee of the Regierungspräsidium Freiburg (license numbers G16-152 and G19-22).

## Decision letter and Author response
Decision letter https://doi.org/10.7554/eLife.79471.sa1
Author response https://doi.org/10.7554/eLife.79471.sa2

---

## Additional files

### Supplementary files
• MDAR checklist

### Data availability
Single-unit data and analysis code of this manuscript are available as the following data set: Sauer JF, Bartos M | 2022 | Data and analysis code for Sauer & Bartos, 2022 | https://zenodo.org/record/7115771 | Zenodo, https://doi.org/10.5281/zenodo.7115771.

The following dataset was generated:

| Author(s) | Year | Dataset title | Dataset URL | Database and Identifier |
|---|---|---|---|---|
| Sauer JF, Bartos M | 2022 | Data and analysis code for Sauer & Bartos, 2022 | https://zenodo.org/record/7115771 | Zenodo, 10.5281/zenodo.7115771 |

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
