## [Editor Report]

This article explores prefrontal cortex circuit function in mice lacking *disrupted-in-schizophrenia-1* (Disc1-mutant mice). a risk factor for psychiatric disorders. The data show specific impairment in the function of specific cortical interneuron populations, such as fast-spiking interneurons. Furthermore, it also showed a decreased spike transmission efficacy at local pyramidal cell–interneuron connections in vivo. The impaired interneuron function also resulted in altered activation of pyramidal cell assemblies.

---

## [Decision Letter]

**Decision letter after peer review:**

Thank you for submitting your article "Disrupted-in-Schizophrenia-1 is required for pyramidal cell-interneuron communication and network dynamics in the prefrontal cortex" for consideration by *eLife*. Your article has been reviewed by 3 peer reviewers, and the evaluation has been overseen by a Reviewing Editor and Laura Colgin as the Senior Editor. The following individual involved in review of your submission has agreed to reveal their identity: Duda Kvitsiani (Reviewer #1).

Essential revisions:

(1) Electrophysiological analyses should improve, specifically on: neurons average firing frequencies, opto-tagging, spike cross-correlations and cell assembly, following the suggestions made by reviewer 1 and in part by reviewer 2.

(2) Consider the possibility of performing an alternative analysis of single unit spiking in relation to γ network oscillations and possibly evaluate the spiking activity to other network frequency bands, as suggested by reviewer 2.

(3) Statistical tests for some data should improve, as explained by reviewers 2 and 3, eg, statistical comparisons between genotypes should be performed using each mouse as an independent measurement.

(4) Discuss the consequences of "assembly activations" and the potential mechanisms underlying the observed reduced strength of synaptic connections between excitatory neurons and fast spiking neurons, as suggested by reviewer 3.

*Reviewer #1 (Recommendations for the authors):*

Line 52. It was not clear from the description of the behavioral test (delayed non match to place paradigm) how many trials on average and s.t.d. was collected from two mouse genotypes.

Line 55. Authors should report how many cells were recorded from Disc1 and control mice.

Line 58. t(176) = 3.686 stands for mean firing rate of 176 neurons? It was not clear what was the numbers meant.

Line 94. "(χ2 -test, data not shown)". I suggest to either show the data or remove the statement.

Line 213. "a and asymmetry" a should be removed.

Line 395. There are two "the".

Figure 4.b What are the axis labels and color scale?

*Reviewer #2 (Recommendations for the authors):*

1. Histological verification of the position of probes and tetrodes would strengthen the conclusions. It is stated in the methods but not shown.

2. Figure 1e is misleading. Spikes observed in a single trace do not necessarily originate in a single neuron. Therefore, the increase can not be attributed to one unit. Is the isolation performed on the timestamps of isolated units or on the traces as suggested by the figure?

3. Power spectral density seems to be normalized although it is unclear how. Variations in the power of low frequency bands suggest contamination by slow drift in the recorded signal. Strong filtering using 0.5 or 1 Hz as lower cut-off might solve it.

4. Discussion (lines 185-192): this paragraph seems contradictory. Even if PVIs can accomplish efficient inhibition of PYR spiking activity in DISC-1 mPFC, this would only occur when the PVI fires an action potential, meaning at lower frequency. Therefore, there would still be an increase of PYR spiking rate.

*Reviewer #3 (Recommendations for the authors):*

At present, there are some major issues with this manuscript concerning its relationship with previous work, and the statistical analysis of the data:

1. Reduced firing rates of fast-spiking interneurons in the Nucleus accumbens of DISC1 mice have previously been reported (pmid 34143365). Furthermore, reduced input from excitatory neurons as a reason for GABAergic dysfunction has previously been implicated in schizophrenia pathophysiology (e.g. pmid 27244370). While the present study uses more advanced recording and analysis methods, the progress made compared to these previous studies should be delineated more clearly.

2. The increased strength of "assembly activations" is one of the major novelties of the present manuscript, yet it is reported as a raw value without any interpretation. What is the consequence of this increased "assembly strength" for microcircuit dysfunction and for cognitive deficits? Are "stronger assemblies" expected to be more or less plastic, more or less responsive, more or less stable across time? How are these changes expected to contribute to cognitive disturbances?

3. The present study proposes that reduced input from excitatory neurons contributes to GABAergic dysfunction, yet it does not explain the mechanism underlying this reduced excitatory drive. Previous work has suggested that disruption of NMDARs (pmid 19915563) or calcium channels (pmid 33863995) may contribute to reduced excitability of GABAergic interneurons in schizophrenia models. In the absence of any data exploring the mechanism, at the very least, these previous findings should be discussed in the present manuscript.

4. The data are of nested nature (animals-sessions-cells/units) yet the analysis only uses cells as statistical units. Ideally, the statistical assessment should account for the hierarchical nature of the data. At a minimum, the authors should ensure that the variability across animals and sessions is small enough to warrant using cells as the statistical unit, and per-animal averages should be shown in the figures so that the readers can judge the variability across animals for themselves.

---

## [Author Response]

Essential revisions:1) Electrophysiological analyses should improve, specifically on: neurons average firing frequencies, opto-tagging, spike cross-correlations and cell assembly, following the suggestions made by reviewer 1 and in part by reviewer 2.

We revised our analyses following the reviewers’ suggestions. In particular, we tested neurons with low/high firing rates as proposed by reviewer 1, performed additional control analyses to strengthen our opto-tagging approach including a systematic comparison of the firing rates and waveform features of pyramidal cells (PYRs), interneurons (INT) and optogentically identified parvalbumin-positive interneurons (PVIs), and an analysis of the kinetics of firing rate changes in opto-tagged PVIs upon laser onset. These analyses confirmed our original observations. In addition, we performed new analysis on INT-INT synchronization and INT spike-triggered γ amplitudes (as requested by reviewer 1 #5 and 6).

2) Consider the possibility of performing an alternative analysis of single unit spiking in relation to γ network oscillations and possibly evaluate the spiking activity to other network frequency bands, as suggested by reviewer 2.

We have completely revised our phase-coupling analysis. Following the reviewer’s suggestion, we obtained surrogate data for each unit by shuffling the inter-spike intervals (500 iterations), and tested for significant phase-locking as crossings of the 95^th^ percentile of the random distribution in two consecutive bins. To include additional frequencies, we performed this analysis for the low (20-40 Hz) and high γ bands (60-90 Hz) for both INTs and PYRs. In addition, to better account for the effect of firing rates on our measure of phase coupling depth (PPC), we used random subsampling of 250 spikes (100 iterations), computed PPC in each iteration, and averaged over the PPC estimates per cell. Finally, the results of all neurons (PYR and INT separately) were then averaged for each mouse.

Consistent with our original analysis, we found a reduced proportion of high γ-coupled INTs and no significant effects for low γ oscillations or for the phase-coupling of PYRs to either low or high γ bands. These results are now shown in the new Figure 4 and the new Figure 4 —figure supplements 1 and 2. For details of our revised analysis, please see our response to reviewer 2 #3.

3) Statistical tests for some data should improve, as explained by reviewers 2 and 3, eg, statistical comparisons between genotypes should be performed using each mouse as an independent measurement.

We performed additional recordings, yielding a final data set of 9 Disc1 and 7 control mice, and recomputed the main results of this study based on mouse averages. Using this approach, we could replicate the major findings of the original cell-by-cell analysis, including reduced INT firing rates, decreased spike transmission probability, lower phase-coupling of INT spikes to high γ oscillations, and reduced strength of assembly activations in Disc1-mutant mice. Our original finding of enhanced synchronization of PYRs could not be reproduced on the level of individual mice, which is clearly reported in the revised manuscript.

The analyses based on opto-tagging did not allow group statistics at the mouse level due to the low number of experiments (n=3 mice each). However, given that the optotagging experiments are only confirmatory in nature and that additional optotagging experiments would result in extensive additional experimental work, we report the results of these analyses as comparisons between neurons or connected pairs. This is clearly stated in the revised manuscript. We hope that the editors and reviewers concur with our decision.

4) Discuss the consequences of "assembly activations" and the potential mechanisms underlying the observed reduced strength of synaptic connections between excitatory neurons and fast spiking neurons, as suggested by reviewer 3.

We included a discussion of the potential functional implications of enhanced assembly activation strength in the discussion. In brief, we propose that assemblies in Disc1-mutant mice might be more stable over time, which could potentially impact the balance between stable encoding of memory and the flexible acquisition of new memories. We moreover discuss potential causes of the reduced spike train transmission, including changes in axonal wiring, synaptic efficacy and excitability of INTs.

Reviewer #1 (Recommendations for the authors):Line 52. It was not clear from the description of the behavioral test (delayed non match to place paradigm) how many trials on average and s.t.d. was collected from two mouse genotypes.

Working memory testing was performed over three days with 3-10 trials at 15 s delay following the sample run (Days 1 and 2: 3 runs, Day 3: 10 runs).

Unfortunately, upon accessing the data to double-check the requested information on the number of trials, we noted that the original videos used to analyze working memory of four Disc1-mutant mice are lost. These seem not retrievable from our backup system. In the remaining sample, we still observed a trend towards reduced working memory performance of Disc1-mutant mice (76 ± 6 % correct) compared to control mice (83 ± 10 % correct). This result was not significant (t = -1.457, p=0.167, unpaired t-test), presumably due to the insufficient number of mice. Since the working memory deficit of Disc1-mutant mice has been previously published (Koike et al., 2006, Proc. Natl. Acad. Sci. USA 103:3693-3697, Kvajo et al., Proc. Natl. Acad. Sci. USA 105:7076-7081) and is not an essential part of our investigation of the network properties in these animals, we have removed the original Figure 1—figure supplement 1 from the manuscript. We hope that the reviewer agrees with this decision.

Line 55. Authors should report how many cells were recorded from Disc1 and control mice.

We recorded from a total of 720 and 613 pyramidal neurons and 104 and 79 putative INTs in Disc1 and control mice, respectively. We have included this missing information in the main text in line 65.

Line 58. t(176) = 3.686 stands for mean firing rate of 176 neurons? It was not clear what was the numbers meant.

We originally reported the degrees of freedom in brackets. In the revised manuscript, we changed the report of statistical results and now state ‘test statistic, p-value, n, used test’, thereby following the Journal’s guidelines.

Line 94. "(χ2 -test, data not shown)". I suggest to either show the data or remove the statement.

We have reanalyzed connection probability based on mouse averages as requested by reviewer 1 (#1) and reviewer 3 (#4). The new analysis showed no significant difference in connection probability, which is now described in line 105 ff.

Line 213. "a and asymmetry" a should be removed.

In fact, ‘a’ refers to the duration. It is shown in the inset to panel a to emphasize how waveform duration was measured. We have changed the figure legend to ‘[…] based on duration (a) and asymmetry index […]’ for clarification.

Line 395. There are two "the".

We thank the reviewer for spotting this typo. We have removed it.

Figure 4.b What are the axis labels and color scale?

The axes of the covariance matrix indicate the number of PYRs. We have added according x/y labels and show the color scale (covariance) in the revised Figure (which is now Figure 5).

Reviewer #2 (Recommendations for the authors):1. Histological verification of the position of probes and tetrodes would strengthen the conclusions. It is stated in the methods but not shown.

We have added the missing information in the new Figure 1 —figure supplement 1.

2. Figure 1e is misleading. Spikes observed in a single trace do not necessarily originate in a single neuron. Therefore, the increase can not be attributed to one unit. Is the isolation performed on the timestamps of isolated units or on the traces as suggested by the figure?

We apologize for the misleading presentation of the data. Unit isolation was performed using all available channels, the display of the channel with the largest waveform was merely chosen for illustrative purposes. This is clearly stated in the revised figure legend. Following # 2 by reviewer 1 we moreover revised the display of the optotagging data and now show single channels of all optogenetically identified PVIs in the revised Figure 1f.

3. Power spectral density seems to be normalized although it is unclear how. Variations in the power of low frequency bands suggest contamination by slow drift in the recorded signal. Strong filtering using 0.5 or 1 Hz as lower cut-off might solve it.

We normalized the data by z scoring but applied no additional normalization (such as 1/f or 1/f^2^). This is now described more clearly in the Methods section in line 490. Variation in the lower frequency band was indeed observed to some extent. However, we found no difference in theta (6-12 Hz,) power between Disc1-mutant and control mice (see the new Figure 4a,b), suggesting that the lower frequencies are unaffected in these mice.

4. Discussion (lines 185-192): this paragraph seems contradictory. Even if PVIs can accomplish efficient inhibition of PYR spiking activity in DISC-1 mPFC, this would only occur when the PVI fires an action potential, meaning at lower frequency. Therefore, there would still be an increase of PYR spiking rate.

This is true. However, it is also conceivable that given the strong convergence of many PVIs onto a given PYRs the postsynaptic cell will receive sufficient inhibition at any given time point such that the average firing rate might not change. However, additional explanations can be considered such as reduced excitatory drive among PYRs (both local and long-range), which could compensate the lower levels of inhibition, or altered properties of non-fast spiking INT types (e.g., enhanced inhibition by somatostatin-positive neurons). These possibilities should be tested in future and are now discussed more explicitly in the revised discussion in line 272.

Reviewer #3 (Recommendations for the authors):At present, there are some major issues with this manuscript concerning its relationship with previous work, and the statistical analysis of the data:1. Reduced firing rates of fast-spiking interneurons in the Nucleus accumbens of DISC1 mice have previously been reported (pmid 34143365). Furthermore, reduced input from excitatory neurons as a reason for GABAergic dysfunction has previously been implicated in schizophrenia pathophysiology (e.g. pmid 27244370). While the present study uses more advanced recording and analysis methods, the progress made compared to these previous studies should be delineated more clearly.

We thank the reviewer for pointing out these highly relevant publications. We have revised our introduction to better reflect the existing literature, including the publications suggested by the reviewer, on impaired PVI activity and potential PYR-dependent mechanisms underlying PVI impairments in the revised manuscript in line 41 ff.

2. The increased strength of "assembly activations" is one of the major novelties of the present manuscript, yet it is reported as a raw value without any interpretation. What is the consequence of this increased "assembly strength" for microcircuit dysfunction and for cognitive deficits? Are "stronger assemblies" expected to be more or less plastic, more or less responsive, more or less stable across time? How are these changes expected to contribute to cognitive disturbances?

We fully agree that further interpretation of these results is warranted. We propose that increased assembly activation strength might lead to enhanced stability of the assemblies over time. This is based on the conjecture that Hebbian plasticity mechanisms are crucial for the emergence and maintenance of assemblies. Synaptic plasticity has indeed been demonstrated to be enhanced in Disc1-mutant mice (at least for layer 2/3-to-layer 5 connections in the mPFC) and might contribute to the observed stronger assembly activation. Stronger assembly activations could thus stabilize the assemblies over time, which might favor stable encoding of memory content over flexible encoding of new information in the circuitry. This might, in turn, relate to impaired working memory performance of Disc1-mutant mice (Koike et al., 2006, Proc. Natl. Acad. Sci. USA 103:3693-3697, Kvajo et al., Proc. Natl. Acad. Sci. USA 105:7076-7081). These considerations are discussed in the revised version of the manuscript starting in line 296.

3. The present study proposes that reduced input from excitatory neurons contributes to GABAergic dysfunction, yet it does not explain the mechanism underlying this reduced excitatory drive. Previous work has suggested that disruption of NMDARs (pmid 19915563) or calcium channels (pmid 33863995) may contribute to reduced excitability of GABAergic interneurons in schizophrenia models. In the absence of any data exploring the mechanism, at the very least, these previous findings should be discussed in the present manuscript.

Following the reviewer’s suggestion we have revised our discussion, which now includes a discussion of the potential mechanisms of reduced spike transmission to INTs (line 238 ff.). These include altered axonal wiring, reduced efficacy of excitatory synapses at PYR-INT connections, and lower excitability of INTs in Disc1-mutant mice.

4. The data are of nested nature (animals-sessions-cells/units) yet the analysis only uses cells as statistical units. Ideally, the statistical assessment should account for the hierarchical nature of the data. At a minimum, the authors should ensure that the variability across animals and sessions is small enough to warrant using cells as the statistical unit, and per-animal averages should be shown in the figures so that the readers can judge the variability across animals for themselves.

Following the reviewer’s suggestion, we have performed additional recordings to be able to reanalyze all major comparisons of the study based on mouse averages (see also reviewer 2 #1). In the majority of cases, the significant differences reported in the original manuscript could be reproduced this way (except for connection probability and PYR synchronization). In case of the analyses with optogentically identified PVIs, the low number of experiments did not allow comparison by mouse averages. Given that the vast majority of our conclusions is based on electrophysiologically identified INTs, with optogenetic identification experiments being only confirmatory in nature, and that performing additional optogentic identification experiments would be very laborious, we report the results of these analyses as comparisons between neurons or connected pairs. This is clearly stated in lines 88, 109 and 141 ff. of the revised manuscript.